# SiO_2_-Ag Composite as a Highly Virucidal Material: A Roadmap that Rapidly Eliminates SARS-CoV-2

**DOI:** 10.3390/nano11030638

**Published:** 2021-03-04

**Authors:** Marcelo Assis, Luiz Gustavo P. Simoes, Guilherme C. Tremiliosi, Dyovani Coelho, Daniel T. Minozzi, Renato I. Santos, Daiane C. B. Vilela, Jeziel Rodrigues do Santos, Lara Kelly Ribeiro, Ieda Lucia Viana Rosa, Lucia Helena Mascaro, Juan Andrés, Elson Longo

**Affiliations:** 1CDMF, LIEC, Federal University of São Carlos—(UFSCar), 13565-905 São Carlos, SP, Brazil; marcelostassis@gmail.com (M.A.); dyovani@gmail.com (D.C.); prof.jeziel@gmail.com (J.R.d.S.); larakribeiro@gmail.com (L.K.R.); ilvrosa@ufscar.br (I.L.V.R.); lmascaro@ufscar.br (L.H.M.); elson.liec@gmail.com (E.L.); 2Department of Physical and Analytical Chemistry, University Jaume I (UJI), 12071 Castellon, Spain; 3Nanox Tecnologia S/A, 13562-400 São Carlos, SP, Brazil; gustavo@nanox.com.br (L.G.P.S.); guilherme@nanox.com.br (G.C.T.); daniel@nanox.com.br (D.T.M.); renato.santos@nanox.com.br (R.I.S.); microbiologia@nanox.com.br (D.C.B.V.)

**Keywords:** COVID-19, virus elimination, antiviral surfaces, SiO_2_-Ag composite, ethyl vinyl acetate, surface plasmon resonance effect

## Abstract

COVID-19, as the cause of a global pandemic, has resulted in lockdowns all over the world since early 2020. Both theoretical and experimental efforts are being made to find an effective treatment to suppress the virus, constituting the forefront of current global safety concerns and a significant burden on global economies. The development of innovative materials able to prevent the transmission, spread, and entry of COVID-19 pathogens into the human body is currently in the spotlight. The synthesis of these materials is, therefore, gaining momentum, as methods providing nontoxic and environmentally friendly procedures are in high demand. Here, a highly virucidal material constructed from SiO_2_-Ag composite immobilized in a polymeric matrix (ethyl vinyl acetate) is presented. The experimental results indicated that the as-fabricated samples exhibited high antibacterial activity towards *Escherichia coli* (*E. coli*) and *Staphylococcus aureus* (*S. aureus*) as well as towards SARS-CoV-2. Based on the present results and radical scavenger experiments, we propose a possible mechanism to explain the enhancement of the biocidal activity. In the presence of O_2_ and H_2_O, the plasmon-assisted surface mechanism is the major reaction channel generating reactive oxygen species (ROS). We believe that the present strategy based on the plasmonic effect would be a significant contribution to the design and preparation of efficient biocidal materials. This fundamental research is a precedent for the design and application of adequate technology to the next-generation of antiviral surfaces to combat SARS-CoV-2.

## 1. Introduction

The current worldwide public health and economic crisis resulting from COVID-19 has become a critical problem [1]. At present, there are no vaccines or antiviral drugs available for the prevention or treatment of COVID-19 infections. Currently, many different antiviral agents, including repurposed drugs, are under testing in clinical trials to assess their efficacy, but the quest for an effective treatment against COVID-19 is still ongoing [2,3,4,5]; therefore, it is essential to explore any other effective intervention strategies that may reduce the mortality and morbidity rates of the disease. Some excellent reviews of therapeutics and tools that inactivate SARS-CoV-2 have been published [6,7,8].

SARS-CoV-2 spreads mainly via human fluids, and individuals may acquire the virus after touching different contaminated surfaces [9]. It is known that SARS-CoV-2 remains viable on solids for extended periods (for up to 1 week on hard surfaces such as glass and stainless steel) [10,11]. Consequently, not only is the identification of materials capable of killing viruses by contact and having low cytotoxicity clearly a high priority for all scientists around the world, but the detection of new and effective materials to decontaminate surfaces is also of great concern [12,13,14]. Given the significance of surface and air contamination in the spread of the virus, attention should also be paid to the development of biocidal (virus, bacteria, fungus) materials against the spread of contamination facilitated by frequently touched surfaces, such as protecting hospital environments and the surfaces of biomedical devices, along with decontamination equipment and technologies [6,15,16,17,18].

In this scenario, metals, semiconductors, and inorganic materials are gaining increased attention as broad-spectrum antiviral agents to protect surfaces and packaging, thus preventing new infections in humans [19]. Very recently, Ghaffari et al. [20] discussed efforts to deploy nanotechnology, biomaterials, and stem cells in each step of the fight against SARS-CoV-2, while Basak and Packirisamy [21] have discussed several nanotechnological strategies that can be used as antiviral coatings to inhibit viral transmission by preventing viral entry into host cells. In this context, metal oxide nanoparticles and their composites were established as potent antibacterial agents due to the induced generation of reactive oxygen species (ROS) and the subsequent oxidative stress [22,23]. They can still enter the microorganism’s membranes, reacting with the existing phosphate and sulfate groups, impairing their functioning, and consequently leading to the microorganism’s death [24,25]. ROS can still inhibit the replication activities of DNA/plasmid and some protein enzymes, due to their interaction with phosphate/sulfate groups or even due to genetic changes [25,26]. All of these results, in combination with the permeability of ROS under the cell membrane, can affect the expression of proteins essential for the correct functioning of microorganisms, as well as their replication [24,27,28,29,30,31].

In particular, silver (Ag) is a widely known element for its antimicrobial properties and has been used in colloidal silver compounds or as adsorbed particles in a colloidal carrier [32]. In addition, Ag nanoparticles (Ag NPs) display the antimicrobial properties of bulk Ag, with a significant reduction in the toxic effects observed with Ag cations [33,34,35]. The antimicrobial effects of Ag NPs are accomplished by a unique physiochemical property to generate more efficient contact with microorganisms and enhance interactions with microbial proteins [36]. Ag NPs present excellent activity against many kinds of bacteria [37,38,39,40,41,42] and are capable of disrupting the mitochondrial respiratory chain, leading to the production of ROS [43], and have also demonstrated promising antifungal [44,45] and antiviral capabilities against viruses such as HIV, Tacaribe virus, and several respiratory pathogens, including adenovirus, parainfluenza, and influenza (H3N2) [31,46,47,48,49]. Specifically with regard to antiviral activities, AgNPs are thought to inhibit the entry of the virus into cells due to the binding of envelope proteins, such as glycoprotein gp120, which prevents CD4-dependent virion binding, fusion, and infectivity [31]. In most cases, Ag NPs present the disadvantage of their tendency to agglomerate, leading to a loss of effectiveness. In recent years, the construction of Ag metal/semiconductor composite materials has been identified as a promising strategy for responding to the above problems. Therefore, the strong surface plasmon resonance (SPR) adsorption and high electron trapping ability of Ag NPs are beneficial for promoting the charge transmission bridge [29,50,51,52,53,54,55]. This modification of Ag NPs by light establishes a coulombic restoring force and prompts a charge density, and they are frequently used in plasmonic composites.

Among the large number of metal/semiconductor composites, SiO_2_-Ag has attracted considerable attention due to its excellent properties, because SiO_2_ is thermally stable and highly bioactive, and could not only prevent the agglomeration of particles and enhance the surface hydrophilicity but also further improve their stability [56,57,58,59,60,61,62,63,64,65]. Recently, it has been demonstrated that mesoporous silica nanoparticle/Ag composite presents great potential as a candidate for the development of products aiming to prevent the spread of drug-resistant pathogens [66,67].

An important feature of such materials is the combination of positive properties of the polymer matrix, such as lightness, flexibility, and ease of production, as well as the ability to modify the properties of the material. However, the Ag NPs hosted in SiO_2_ have certain drawbacks in relation to their stability. This situation has spurred the study of alternatives allowing viability for technological applications such as their immobilization in a physical support such as a polymer matrix [68,69] and additional reducing agents or capping agents [70]. Polymers displaying antimicrobial properties are the subject of significant attention for their potential technical and medical applications [71,72,73,74]. One of the most promising types of such materials is based on a SiO_2_-Ag composite immobilized in a polymeric matrix, which has properties that are individually not achievable for each of the components.

Very recently, our research group presented the development and manufacture of materials with anti-SARS-CoV-2 activity, generating potentially safe alternatives for their application, preventing viral proliferation and transmission [27]. Herein, we report the results of our studies on the structure and properties of SiO_2_-Ag composite immobilized in a polymeric matrix (ethyl vinyl acetate, EVA). Their antibacterial activity towards *Escherichia coli* (*E. coli*) and *Staphylococcus aureus* (*S. aureus*) as well as towards SARS-CoV-2 have been investigated. The synthesized materials were characterized by X-ray diffraction (XRD), field emission scanning electron microscopy (FE-SEM), and micro-Raman spectroscopy. Moreover, their optical properties were investigated by using ultraviolet−visible (UV−vis) spectroscopy. In addition, first-principles calculations within the framework of Density functional theory (DFT) were employed to obtain atomic-level information on the geometry and electronic structure, local bonding of the SiO_2_ model, and their interaction with $O_{2}$ and $H_{2}O$. Furthermore, we explored the application of the samples for the photocatalytic activity in the degradation of Rhodamine B (RhB) and trapping experiments were carried out to understand the radical scavenging behavior. The broad spectrum of interesting properties displayed by such materials present opportunities for a multitude of biomedical applications.

## 2. Materials and Methods

Synthesis Ag NPs: Briefly, silver nitrate (850 mg, AgNO_3_, Cennabras (Guarulhos, Brazil), 99.8%) was dissolved in 100 mL of deionized water at 90 °C and stirred until complete dissolution. Subsequently, 1.0 mL of sodium citrate (C_6_H_5_Na_3_O_7_, Sigma-Aldrich (St. Louis, MO, USA), 98%) diluted in water (1% (wt/wt)) was added and the transparent solution converted to a yellowish-green colloid, which indicated the formation of Ag NPs. After 1 h, the colloidal dispersion was mixed with 11g of amorphous SiO_2_ (Sigma-Aldrich, St. Louis, MO, USA) and dried at 125 °C in a conventional oven.

Preparation of EVA-SiO_2_-Ag Composite: EVA 3019, melt index 2.5 g/10 min, was purchased from Braskem (Guarulhos, Brazil). EVA-SiO_2_-Ag masterbatch was prepared by incorporation in the molten state processing of the SiO_2_-Ag into the EVA using a co-rotational twin-screw extruder Plastic AX, Brazil. Mineral oil was used as a compatibilizer agent to prevent agglomeration and to provide uniform distribution of the SiO_2_-Ag into the EVA matrix. Then, 1% in weight of mineral oil (USP Grade, Anastacio Chemistry, São Paulo, Brazil) was firstly dispersed in the polymer by drumming for 20 min at 15 Hz. Subsequently, 10% in weight of SiO_2_-Ag was added to the mixing drum and the process was maintained for an additional time of 20 min. The processing extrusion temperature was 140 °C. To examine the antimicrobial properties of a typical application product, EVA-SiO_2_-Ag composite samples were produced using a thermoplastic injection-molding process. Test samples were produced by dry-blending the EVA polymer with the required amount of masterbatch containing the SiO_2_-Ag additive, which was followed by injection-molding. The samples were 50 by 50 by 1.5 mm and contained the melt-blended EVA composite masterbatch (10% (wt/wt), corresponding to approximately 50 ppm Ag). 

Characterizations: The samples were structurally characterized by XRD using a D/Max-2500PC diffractometer (Rigaku, Tokyo, Japan) with Cu *Kα* radiation (*λ* = 1.5406 Å) in the 2*θ* range of 10–50° and a scanning speed of 1° min^-1^. Furthermore, micro-Raman spectra were recorded using the iHR550 spectrometer (Horiba Jobin-Yvon, Kyoto, Japan) coupled with a Silicon CCD detector and an argon-ion laser (Melles Griot, Rochester, NY USA), which operated at 514.5 nm with a maximum power of 200 mW; moreover, a fiber optic microscope was also employed. Fourier-transform infrared spectroscopy (FT-IR, Bruker Vector 22 FTIR, Billerica, MA, USA) of the samples was recorded at 400–4000 cm^−1^. UV–vis diffuse reflectance measurements were obtained using a Varian Cary spectrometer model 5G in diffuse reflectance mode, with a wavelength range of 2000 to 250 nm and a scan speed of 300 nm min^−1^. An analysis of the thermal stability of samples was conducted on a thermogravimetric (TG/DTA) analyzer (NETZSCH—409 Cell) from 30 to 700 °C at a heating rate of 10 °C min^−1^ and in an oxidizing atmosphere (O_2_) with 50 mL min^−1^ flux. The morphologies, textures, and sizes of the samples were observed with a FE-SEM, which operated at 2 kV (Supra 35-VP, Carl Zeiss, Jena, Germany). A Jem-2100 LaB6 (Jeol, Tokyo, Japan) high-resolution transmission electron microscope (HR-TEM) with an accelerating voltage of 200 kV coupled with an INCA Energy TEM 200 (Oxford, Abingdon, UK) was used to obtain larger magnifications and to clearly verify the samples. AFM images were obtained using a Flex-AFM controlled by Easyscan 2 software (Nanosurf, Liestal, Switzerland) in contrast phase mode on an active vibration isolation table (model TS-150, Table Stable LTD^®^). The cantilever used for image acquisition was the silicon Tap190G (resonant frequency 190 kHz, force constant 48 N/m, Budget Sensors) in setpoint of 50%.

Bactericidal Tests: The bactericidal activity towards *E. coli* and *S. aureus* of the pure polymer and the composite with SiO_2_-Ag was evaluated according to the standard test methodology described in ISO 22196—Measurement of antibacterial activity on plastics and other non-porous surfaces [75], carried out in Nanox’s microbiology laboratory. A 100-μL volume of the bacterial solution (in a concentration of 105 CFU/mL) was inoculated in triplicate over the surface of the samples. The inoculum was then covered with a sterile plastic film which was gently pressed to be distributed throughout the sample area. Samples were incubated in a bacteriological oven at 36 °C for 24 h at 90% humidity. After incubation, the inoculum was recovered with 10 mL of SCDLP broth followed by serial dilution to 10^−4^ in PBS buffer. One mL of each dilution was plated with Standard Count Agar by Pour Plate. After solidification of the culture medium, the Petri dishes were incubated in the inverted position in a bacteriological oven at 36 °C for 24 h. The logarithmic reduction and percentage reduction by the CFU/mL count were then determined by the following equation:(1)$$R=\left(U_{t}-U_{0}\right)-\left(A_{t}-U_{0}\right)=U_{t}-A_{t}$$
where *R* is the antibacterial activity; *U*_0_ is the average of the common logarithm of the number of viable bacteria, in cells/cm^2^, recovered from the untreated test specimens immediately after inoculation; *U_t_* is the average of the common logarithm of the number of viable bacteria, in cells/cm^2^, recovered from the untreated test specimens after 24 h, and *A_t_* is the average of the common logarithm of the number of viable bacteria, in cells/cm^2^, recovered from the treated test specimens after 24 h.

Antiviral Tests: The antiviral activity of the pure polymer and the composite with SiO_2_-Ag was evaluated by adapting the standard model ISO 21702—Measures of antiviral activity on plastics and other non-porous surfaces [76] and the method used by Tremiliosi et al. [27]. The tests were carried out in a NB3 (biosafety level 3) laboratory at the University of São Paulo, following the recommendations of ANVISA. SARS-CoV-2 was inoculated into liquid media; EVA polymer and the EVA-SiO_2_-Ag composite samples were incubated for 2 different time intervals (2 and 10 min). Then, they were seeded in Vero CCL-81 cell cultures. After incubation, the viral genetic material was quantified by quantitative PCR in real time and, based on the control, the ability of each sample to inactivate SARS-CoV-2 was determined.

Reactive Oxygen Species (ROS) Identification: To investigate the active species generated in the photocatalytic RhB (Aldrich, 95%) degradation process over SiO_2_-Ag composite, a trapping experiment was conducted with ascorbic acid (AA), ammonium oxalate (AO), and tert-butyl alcohol (TBA) as the capture agent of hydroxyl radical (${\mathrm{OH}}^{*}$), hole $\left(h^{•}\right)$, and hydroperoxyl radical ($O_{2}H^{*}$), respectively. The trapping experimental procedure was identical to photocatalytic degradation except that an additional capture agent was added each time. In this way, 50 mg of the sample was dispersed in 50 mL of RhB solution (1 × 10^−5^ M), and it was kept in the dark for 30 min at 20 °C, and then 6 visible lamps (Philips TL-D, 15W) were switched on. After 60 min, an aliquot was removed and centrifuged to obtain only the liquid phase. The variations in the standard absorption of RhB (554 nm) were discerned through analysis of absorption spectroscopy in the UV–vis region on a V-660 spectrophotometer (JASCO, Tokyo, Japan).

Computational Method: The calculations were performed with the Gaussian 09 package [77] by using density functional theory (DFT), with the hybrid functional B3LYP and 6-31 ++ G ** basis set. In the Appendix A, the model systems employed in this study are presented. An analysis based on the results of the natural bond orbital (NBO) method (Reed et al.) and the map electrostatic potential (MEP) is employed to investigate the charge transfer process between the SiO_2_ model and O_2_ and H_2_O.

## 3. Results and Discussion

The X-ray diffraction (XRD) measurements are presented in Figure 1. An analysis of the results shows that the sample SiO_2_-Ag has a characteristic peak of amorphous SiO_2_ at around 2*θ* = 22.2° [78,79,80,81]. No additional peak is observed regarding possible Ag phases. For pure EVA, a high crystallinity of the polymer is observed, which is in line with what has been observed in other studies in the literature [82,83]. The SiO_2_-Ag particles were added in a polymeric matrix, EVA, which has the role of carrier. For the EVA-SiO_2_-Ag composite, there is an amorphization of the polymeric structure; that is, the symmetry and periodicity break at long-range. This is due to the high degree of disorder of the distorted tetrahedral clusters of [SiO_4_] present in amorphous SiO_2_ [84], which cause an induction to amorphization of the polymeric EVA chains. As a result of this union, a broad band located at 2*θ* = 19.9° is observed.

In order to complement the results obtained by XRD, micro-Raman measurements were performed, seeking to analyze the degree of order of the samples at short range (Figure 2). For the SiO_2_-Ag sample, a peak of approximately ~240 cm^−1^ is observed, referring to the scissoring of the distorted tetrahedral of the [SiO_4_] clusters [85]. For pure EVA, there are five distinct groups of vibrations in the micro-Raman spectrum [86]. The vibrations in the range 500–700 cm^−1^ correspond to the deformation movements of the acetate groups of the EVA monomers [86,87,88]. A set of peaks related to the C-C stretches of the constituent monomers is observed in the range of 750 to 1250 cm^−1^ [86,89]. The peaks between 1300 and 1500 cm^−1^ were ascribed to the bending and twisting vibrations of the ethylene groups in the monomers of EVA [86,87]. Between 1700 and 1900 cm^−1^, stretches related to C=O bonds are observed [86,90]. At the highest wavelengths, located between 2800 and 3050 cm^−1^, C–H aliphatic stretches of the EVA are observed [87,91]. In contrast to the XRD observations, the composite does not lose its organization at short range; that is, its constituent monomers maintain their degree of structural order. The SiO_2_-Ag mode in the composite can also be observed, indicating good incorporation in the EVA polymer. According to Shen et al., this mode at ~240 cm^−1^ may also refer to vibrations of the Ag-O bonds, which can be formed from the interaction of the O atoms of the carbonyl groups of the EVA with the Ag contained in SiO_2_-Ag [87].

Fourier-transform infrared spectroscopy (FTIR) was performed to analyze changes in the functional groups of the samples and to verify the formation of the composite EVA-SiO_2_-Ag (Figure 3). For SiO_2_-Ag, there is a broad band located near 3400 cm^−1^ and another located at 1627 cm^−1^, both corresponding to the O–H stretching of water and the formed silanol groups (Si–OH), respectively [92,93]. The bands observed at 1100 and 475 cm^−1^, on the other hand, are attributed to symmetrical stretching and bending of Si–O–Si bonds, respectively [92,94,95]. The peaks located at 950 and 845 cm^−1^ indicate the bending of the O–Si–O moiety [80,95]. The low-intensity mode located at 552 cm^−1^ can be attributed to Ag-O stretching, showing the presence of Ag in SiO_2_-Ag [96,97]. For EVA, bands referring to the fingerprint of the polymer are observed at 2954, 2850, 1467, 1243, 874, 707, and 546 cm^−1^, related to the EVA aliphatic groups [98,99,100,101,102]. At 1020 cm^−1^, the bending of the C–O–C bonds is observed [102], and at 1801 and 1739 cm^−1^, the C=O bond stretching refers to two different types of carbonyl groups [101,102], as noted by Poljansek et al. [103]. For the EVA-SiO_2_-Ag composite, changes are observed especially for the stretching of the C=O bonds and throughout the low-wavelength region, where the SiO_2_ vibrational modes appear. This is because EVA monomers interact through ionic and van der Waals forces with SiO_2_ and Ag, shown by the overlap of some vibrational modes of the samples and the appearance of new ones. These findings indicate the interactions between the polymer, at short and long range, with the particles of SiO_2_ and Ag.

The thermogravimetric (TG) and differential thermal analysis (DTA) curves are shown in Figure 4. In the SiO_2_-Ag sample, a small loss of mass (9.3%) is observed at 50 °C, due to the loss of water molecules adsorbed onto the material surface, demonstrating its high thermal stability [104,105]. The degradation of the EVA polymer occurs in two main stages: the first is due to the loss of acetate groups (between 300 and 400 °C) and the second is due to the decomposition of the remaining ethylene groups (between 400 and 650 °C) [106,107]. For the composite, there are no significant differences in the TG/DTA profiles compared to the pure polymer, but a slightly smaller loss of mass occurs for this compound (96.8%) than for the EVA (98.2%). This difference is due to the addition of SiO_2_-Ag in the polymeric structure, which, due to its high thermal stability, does not decompose at higher temperatures.

Figure 5A shows the diffuse reflectance spectra (DRS) of pure EVA and EVA-SiO_2_-Ag, in which light absorption is observed in the range of 685 to 480 nm, attributed to the presence of composite SiO_2_-Ag in the polymer blend. The absorptions on near-infrared wavelengths are ascribed to EVA, where the peaks at 1218, 1440, and 1750 nm are the vibrational modes of the C−H groups in the polymer chain, while the absorption from 1780 to 2000 nm is due to the vinyl acetate group [108,109,110]. The high absorption from 425 nm to ultraviolet wavelengths is attributed to the UV absorber added to the EVA production. The peak at 680 nm is observed for both samples, EVA and EVA-SiO_2_-Ag. The broad absorption due to the presence of SiO_2_-Ag is ascribed to the Ag_2_O nanoparticles in the SiO_2_, as shown in Figure 5B. The same effect was observed by Paul et al. [111] for Ag_2_O nanoparticles growth on TiO_2_ nanorods, in which the composite reduces the bandgap from 2.80 eV (pure TiO_2_) to 1.68 eV. In another report, Deng and Zhu [112] produced nanocomposite spheres of TiO_2_/SiO_2_/Ag/Ag_2_O with a bandgap in the range of 2.19–3.01 eV. Although the Ag_2_O bulk material showed a bandgap from 1.2 to 1.43 eV [113], these values depended on the size of the particle, where the smaller the particle, the higher its bandgap. Here, the bandgap of the SiO_2_-Ag is shown in the inset of Figure 5B, calculated from an indirect interband transition with a value of 1.81 eV. The bandgaps at around 3.03 eV are attributed to the absorption of the UV, which added to EVA production. If a direct electronic transition were considered, only the absorption of the UV absorber would be detected due to the drastic decrease in the diffuse reflectance below 425 nm. The direct transition presents an average energy of approximately 3.26 eV (Figure 5C).

Figure 6 shows the FE-SEM and HR-TEM images for the SiO_2_-Ag sample. It is observed that SiO_2_ microparticles have no defined morphology, due to their degree of amorphization. In addition, on the surface of the larger particles, the deposition of some NPs with greater contrast is observed, indicating the deposition of Ag NPs on the surface of SiO_2_ (Figure 6A,B). To confirm the nature of these deposited NPs, HR-TEM measurements of this sample were performed (Figure 6C,D). As observed in XRD, in the SiO_2_ microparticles, crystalline planes are not observed, confirming that they are amorphous. In addition, smaller crystalline particles associated with a high-contrast surface are observed, as shown in Figure 6D. Fourier-transform (FT) analysis of the crystalline planes of these particles shows that an interplanar distance of 2.35Å was obtained, which is associated with the metallic Ag plane (111) with a cubic structure, according to the card n°44387 [114] in the Inorganic Crystal Structure Database (ICSD), confirming the formation of the SiO_2_-Ag interface. From the EDX analysis of the sample, a Si/Ag ratio (wt/wt) of 25.84 was obtained ( Appendix A).

The 2D AFM images shown in Figure 7 present different characteristics after modification of the EVA with the formation of the SiO_2_-Ag composite. In Figure 7A,B, the height and phase contrast profile for the sample of EVA without the silica-based composite is presented, which provides a surface roughness of 65 nm (root mean square deviation) and a uniform phase contrast with few regions of well-defined contrast. However, the sample EVA-SiO_2_-Ag shows a small surface roughness, 32 nm, and well-defined regions of contrast phase (Figure 7D,E). The 3D AFM images clearly display the roughness differences between the EVA and EVA-SiO_2_-Ag samples, as shown in Figure 7C,F, respectively. Moreover, the dark domains in the contrast phase correspond to the SiO_2_-Ag composite for Figure 7E and present particles of several size scales distributed in the polymeric matrix. Using the image of EVA-SiO_2_-Ag in contrast phase and assuming that all dark domains are SiO_2_-Ag composite, it is possible to verify the presence of 599 particles on the surface in a size scale span from 30 to 385,000 nm^2^. The AFM results are in agreement with the FE-SEM images of the polymer samples. The EVA presents a granular morphology, which is caused by the cure of the polymer blend in its extrusion. After the addition of SiO_2_-Ag, a distribution of particles is observed on the surface of the polymer composite in a broad size scale span, which is in accordance with the AFM images. The broad size distribution of the particles was observed by Hui et al. [115] in the investigation of the low-density polyethylene/ethylene vinyl acetate modification with SiO_2_. Furthermore, the decrease in the surface roughness with the addition of Ag in the polymer matrix was noticed by Filip et al. [116] in their study of polyurethane modified with Ag to produce bionanocomposites.

Once the SiO_2_-Ag particles were successfully incorporated into the EVA, microbiological tests were carried out against *E. coli*, *S. aureus*, and the SARS-CoV-2 virus, due to the high oxidizing power of the Ag NPs combined with the SiO_2_ capacity to produce ROS, which can cause irreversible damage to these microorganisms. The elimination values against *E. coli* and *S. aureus* are shown in Table 1 and the inhibition values against SARS-CoV-2 in Table 2.

For both bacteria, *E. coli* and *S. aureus*, a 99.99% reduction is observed when in contact with the composite after 24 h of incubation. In contrast to the SARS-CoV-2 virus, 99.05% inactivation is observed in 2 min and 99.85% in 10 min for day 1, and 99.26% in 2 min and 99.62% in 10 min for day 2. In both cases, there was no elimination of microorganisms for pure EVA—that is, without the addition of the SiO_2_-Ag composite. This behavior proves the synergistic effect of SiO_2_ microparticles and Ag NPs with EVA.

The microbicidal tests were performed for the EVA-SiO_2_-Ag sample after forced aging by ultraviolet irradiation, following ISO 4892-2: 2013 [117], which aims to reproduce the effects of weathering (temperature, humidity, and/or wetting) that occur when materials are exposed in real-life environments to daylight or daylight filtered through window glass. It was observed that after simulating two years of aging (1200 h of exposure), there is still a 99.950% reduction in the elimination of *S. aureus* and *E. coli*. Thus, the durability defined for the EVA-SiO_2_-Ag was a minimum of two years.

Figure 8 shows the degradation behaviors of the SiO_2_-Ag composite. An analysis of the results shows that the SiO_2_-Ag sample has a photocatalytic efficiency of 23.7% in 60 min (see the degradation kinetics in Appendix A), with a reduction of 0.0, 7.3, and 5.7% in the presence of AA, AO, and TBA, respectively. These findings demonstrate that $h^{•}$, ${\mathrm{OH}}^{*}$, and $O_{2}H^{*}$ are involved in the photodegradation mechanism. These reactive species appear through the formation of $e^{\prime }-h^{•}$ pairs generated in the valence band (VB) and conduction band (CB) [118,119] of the SiO_2_-Ag composite, with subsequent reaction with${\;O}_{2}$ and $H_{2}O$.

SiO_2_ is an n-type semiconductor with a defined electronic structure, bandgap, and position of both CB and VB. Considering the close relation between the photocatalytic and biocidal properties of semiconductors, their activity can be exerted though similar mechanisms. Activation of water ($H_{2}O$) and molecular oxygen ($O_{2}$) are the most important chemical processes involved in both photocatalytic and biocide activities, and the ROS are the key signaling molecules in both processes. As demonstrated by the results of the radical scavenger experiments, SiO_2_ interacts with $H_{2}O$ and $O_{2}$ to provoke the formation of ROS (${\mathrm{OH}}^{*}$ and $O_{2}H^{*}$) [120,121,122,123] and is effective in inhibiting protein adhesion [124,125]. 

First-principles calculations were performed to analyze the interaction of $H_{2}O$ and $O_{2}$ molecules with the SiO_2_ model. We optimize the SiO_2_ model and then the map of the molecular electrostatic potential (MEP) is calculated to investigate the charge transfer process between SiO_2_ and $H_{2}O$ and $O_{2}$ and these results are presented in the Appendix A. The MEP displays the nucleophilic and electrophilic regions where energetically favorable interactions with $H_{2}O$ and $O_{2}$ take place, respectively. At the minima of both interactions, there is an electronic charge of 0.04 e^−^ from H_2_O to SiO_2_ and 0.10 e^−^ from SiO_2_ to O_2_. These events can be considered the early stages of the formation of ${\mathrm{OH}}^{*}$ and $O_{2}H^{*}$. 

The recognized mechanism corresponding to SPR and associated with photoreactivity has not yet been strictly established. In the present study, the proposed photocatalysis and biocidal mechanism of SiO_2_-Ag composites is summarized in Figure 9. The Ag NPs and SiO_2_ particles absorb the incident photons, and the $e^{\prime }$ in the VB in SiO_2_ are excited afterwards. The excited $e^{\prime }\;$move to the CB; at the same time, the same amount of $h^{•}$ is generated in the VB. Because of the higher work function of Ag compared with that of SiO_2_, partially excited $e^{\prime }\;$would transfer from SiO_2_ CB to the surface-loaded Ag NPs, since the Fermi energy level of Ag metal is lower than that of SiO_2_. When the Ag NPs and the SiO_2_ semiconductor come into contact, free electrons migrate from the Fermi level of metallic Ag to the CB of SiO_2_ to reach an equilibrium Fermi state. As a consequence, the whole energy band of the SiO_2_ semiconductor is increased, while that of metallic Ag decreases; this leads to the formation of a depletion layer and an internal electrical field at the interface. The migration of $e^{\prime }\;$away from the depleted region causes the creation of excess positive and negative charges on the Ag NPs’ surface and in the SiO_2_ semiconductor, respectively. Thus, the internal electrical field is directed from Ag NP toward the SiO_2_ semiconductor. Since the SiO_2_-Ag composite is able to absorb the near-ultraviolet to visible light, this helps to absorb more photons and further excite more $e^{\prime }$ within SiO_2_, resulting in the accumulation of more$\;h^{•}$. The $e^{\prime }\;$come up against the $O_{2}$ molecule; meanwhile, $h^{•}$ is quenched by $H_{2}O$ to complete the cycle. Therefore, the biocidal activity of SiO_2_ would be greatly improved if the Ag NPs were anchored onto SiO_2_. To the best of our knowledge, there are still no reports on the utilization of SiO_2_-Ag composite to target SARS-CoV-2. The SiO_2_-Ag composite encapsulated EVA with a narrow bandgap not only efficiently increases the $e^{\prime }$ flow of the SiO_2_ but also largely facilitates the charge separation. The subsequent deposition of Ag NPs promotes electron transfer ability, which leads to higher biocidal activity. Moreover, the contact of Ag NPs with the surface of the semiconductor SiO_2_ can result in an electron-enhanced area in their interface that could effectively facilitate the uptake of electrons and then improve the reduction activity. These reactions can be increased due to the formation of an intense local electric field close to the surface of the Ag NPs (SPR effect) (Figure 9A). At the Ag–semiconductor interface, the number of charge carriers is greater due to the generated electric field, increasing the corresponding separation process (Figure 9B). On the other hand, the interaction with the $e^{\prime }$ in a cluster is represented by the transition of $e^{\prime }$ from occupied to unoccupied states in the band structure. The occupied states are below the Fermi level, and the unoccupied states are mostly above the Fermi level. In the specific case of bacteria, fungi, and viruses, there is an interaction of the region of lower electronic density of the crystal surface with $H_{2}O$. In this interaction, $H_{2}O$ loses an $e^{\prime }$, forming a hydroxyl radical (${\mathrm{OH}}^{*}$) and a proton ($H^{•}$). Simultaneously, an $e^{\prime }$ is transferred to the $O_{2}$ molecule, forming the superoxide anion ($O_{2}^{’}$). This ion, in turn, to maintain the balance of charge and mass, interacts with the $H^{•},$ forming the hydrogen peroxide radical ($O_{2}H^{*}$). The results summarized above are exemplified in Figure 9C.

## 4. Conclusions

The development of new technologies for constructing highly efficient biocidal materials, particularly coating strategies to prevent SARS-CoV-2, is of great significance. Here, a plasmonic SiO_2_-Ag composite immobilized in a polymeric matrix (ethyl vinyl acetate) was successfully prepared and the as-fabricated samples exhibited high antibacterial activity towards *Escherichia coli* (*E. coli*) and *Staphylococcus aureus* (*S. aureus*) as well as towards SARS-CoV-2. The enhancement is mainly due to the SPR effect of the Ag NPs anchored onto the SiO_2_. Considering the close relation between the photocatalytic and biocidal properties of semiconductors, their activity can be exerted though similar mechanisms. The active species trapping experiments suggested that $h^{•}$*,*
${\mathrm{OH}}^{*}$, and $O_{2}H^{*}$ were the main active species for the photocatalytic degradation of RhB and biocidal activity. Given that EVA has high mechanical resistance and stability to water and heat and that the procedure for obtaining the composites is simple and uses low-cost reagents, the SiO_2_-Ag composite has potential advantages for application as a material biocide, and the elimination of SARS-CoV-2. Finally, we propose emerging technologies that have not yet been used for bactericide/virucide purposes but hold great promise and potential for the future engineering of biocidal surfaces. This is the case for the reusable mask manufactured using the EVA-SiO_2_-Ag composite presented here, which has high durability, requiring only the replacement of its filters to have a technology applicable to current demands (Figure 10).

## Figures and Tables

**Figure 1 nanomaterials-11-00638-f001:**
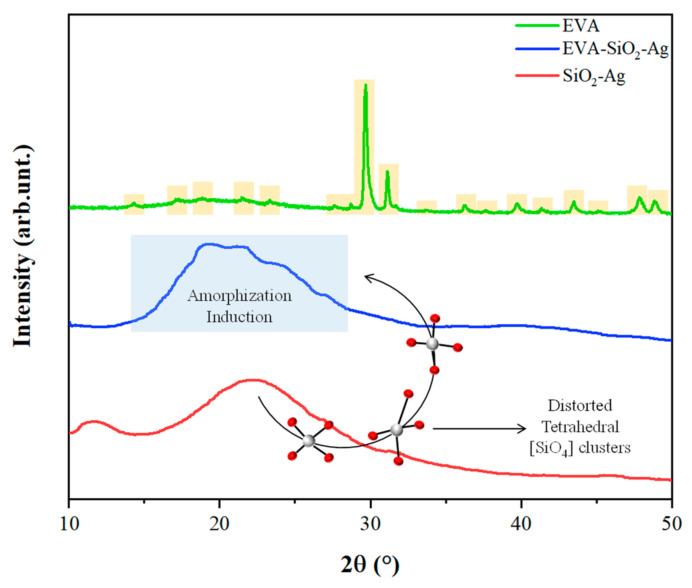
X-ray diffractograms of SiO_2_-Ag, EVA-SiO_2_-Ag, and EVA samples.

**Figure 2 nanomaterials-11-00638-f002:**
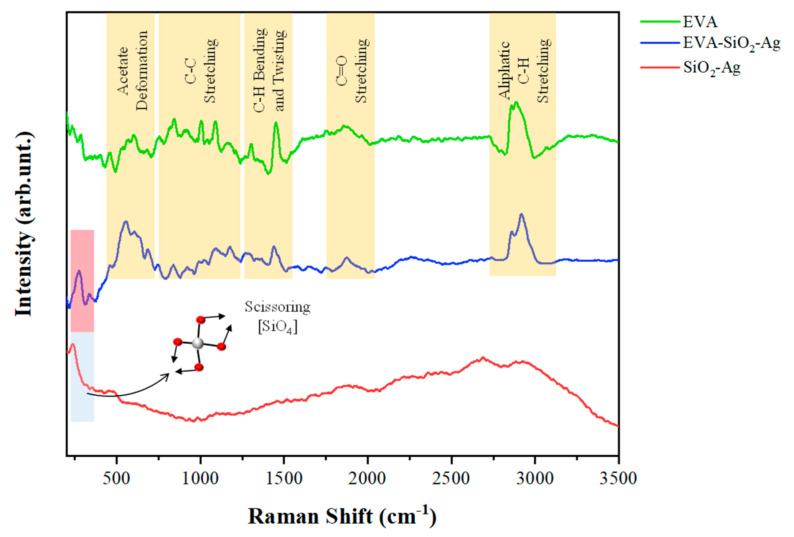
Micro-Raman spectra of SiO_2_-Ag, EVA-SiO_2_-Ag, and EVA samples.

**Figure 3 nanomaterials-11-00638-f003:**
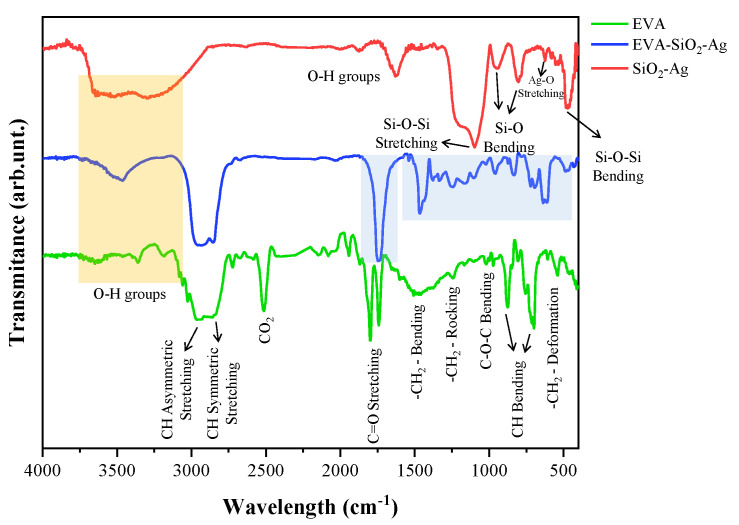
FTIR spectra of SiO_2_-Ag, EVA-SiO_2_-Ag, and EVA samples.

**Figure 4 nanomaterials-11-00638-f004:**
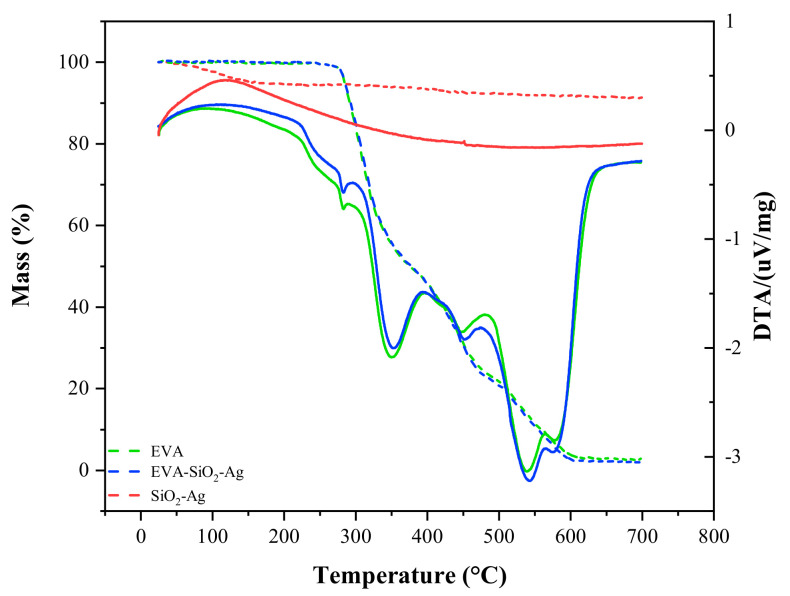
TG/DTA curves of SiO_2_-Ag, EVA-SiO_2_-Ag, and EVA samples.

**Figure 5 nanomaterials-11-00638-f005:**
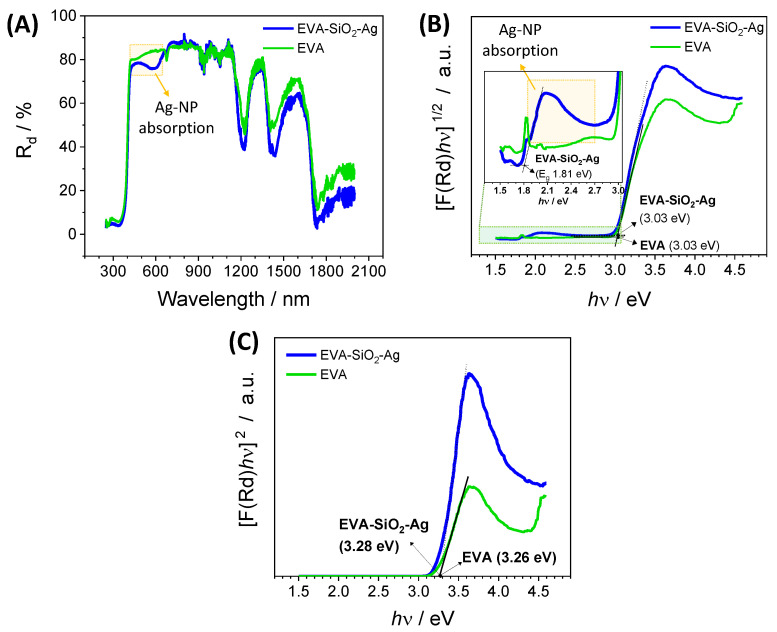
(**A**) Diffuse reflectance spectra, (**B**) indirect interband transition and (**C**) direct interband transition of pure EVA and EVA-SiO_2_-Ag.

**Figure 6 nanomaterials-11-00638-f006:**
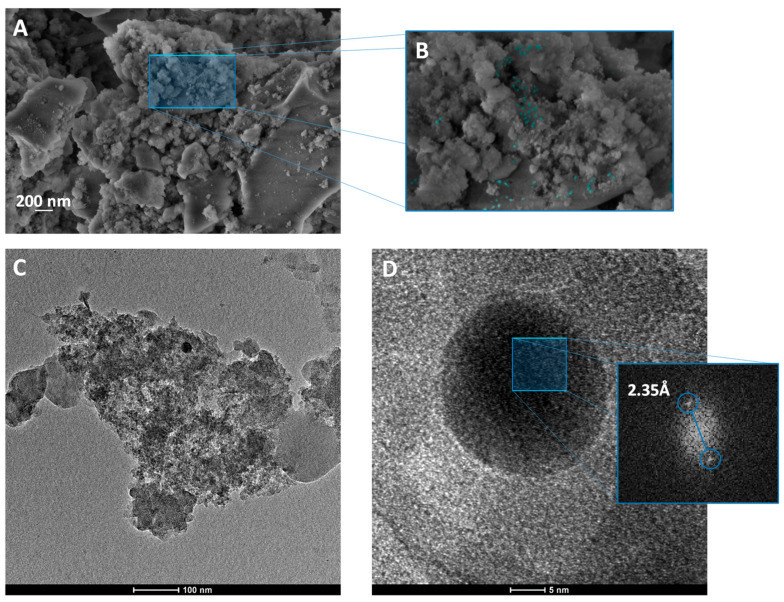
(**A**,**B**) FE-SEM images of SiO2-Ag and (**C**,**D**) TEM and HR-TEM of SiO2-Ag sample.

**Figure 7 nanomaterials-11-00638-f007:**
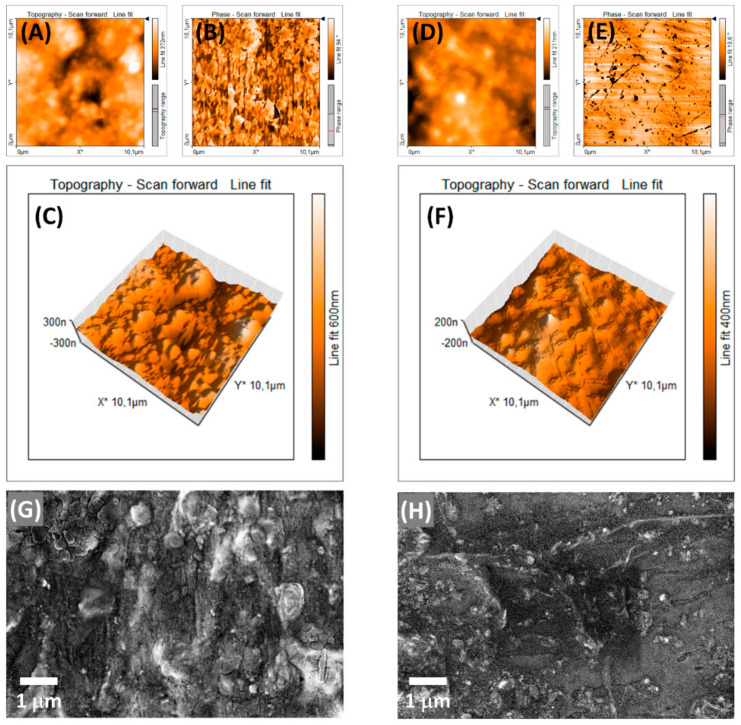
AFM images of (**A**–**C**) EVA and (**D**–**F**) EVA-SiO_2_-Ag samples. SEM images of the (**G**) EVA and (**H**) EVA-SiO_2_-Ag samples.

**Figure 8 nanomaterials-11-00638-f008:**
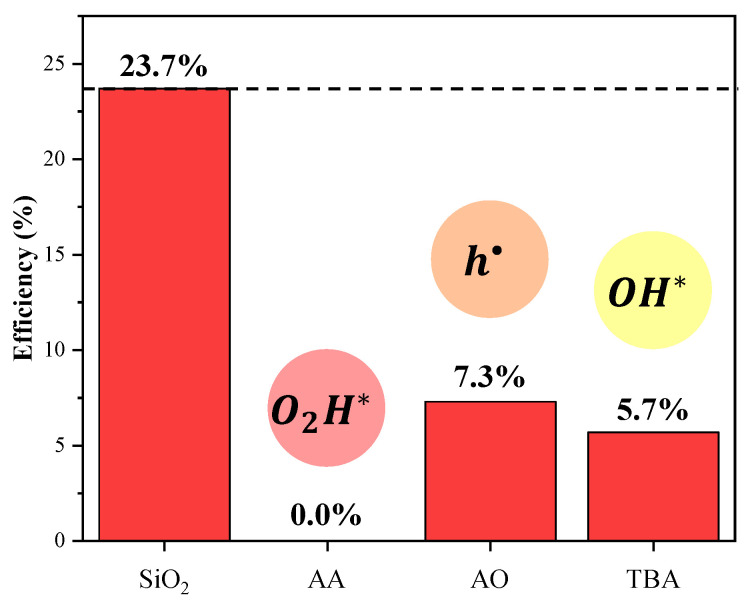
Comparison of photocatalytic degradation of RhB in the presence of different scavengers under visible light irradiation.

**Figure 9 nanomaterials-11-00638-f009:**
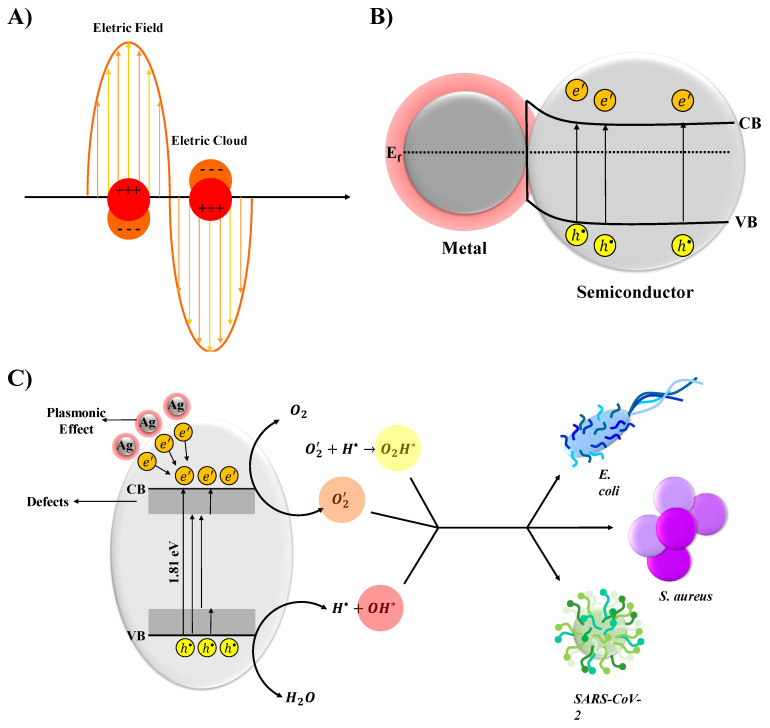
A schematic representation of plasmon-induced hot electrons over SiO2-Ag composite: (**A**) in Ag NP particles; (**B**) in metal semiconductor; and (**C**) proposed mechanism for biocidal activity. (CB and VB represent the conduction band and valence band, respectively.).

**Figure 10 nanomaterials-11-00638-f010:**
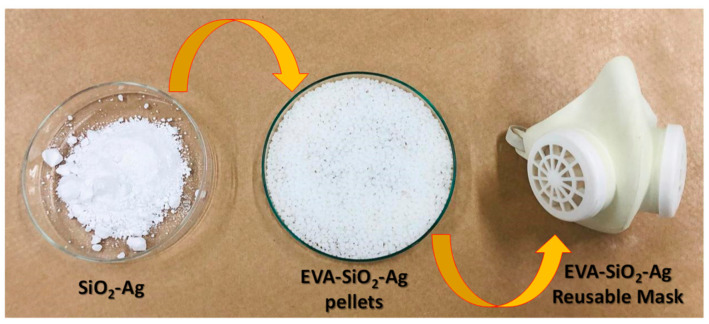
Reusable mask manufactured using the EVA-SiO_2_-Ag composite.

**Table 1 nanomaterials-11-00638-t001:** Results of the efficacy evaluation of biocides incorporated into specimens against *S. aureus* (ATCC 6538) and *E. coli* (ATCC 8739).

	EVA		Eva-SiO_2_-Ag		Reduction in Relation to Control	
	CFU*/test piece (recovery)	Log_10_ of CFU*/test piece (recovery)	CFU*/test piece (recovery)	Log_10_ of CFU*/test piece (recovery)	Reduction in Log_10_	Percentage reduction
*S. aureus*	5.53 × 10^5^	5.74	<1.0 × 10^−1^	<1.0	>4.74	>99.99%
*E. coli*	6.40 × 10^5^	5.80	<1.0 × 10^−1^	<1.0	>4.80	>99.99%

* CFU–colony forming units.

**Table 2 nanomaterials-11-00638-t002:** Copies per mL of SARS-CoV-2 at different times of incubation.

Sample	Incubation Time	Day 1		Day 2	
		Copies/mL (SARS-CoV-2)	Viral Inactivation (%)	Copies/mL (SARS-CoV-2)	Viral Inactivation (%)
EVA	2 min	7.68 × 109	−	3.85 × 108	−
EVA-SiO_2_-Ag	2 min	7.27 × 107	99.05	2.87 × 106	99.26
EVA	10 min	2.21 × 109	−	5.21 × 108	−
EVA-SiO_2_-Ag	10 min	3.28 × 106	99.85	1.98 × 106	99.62

## Data Availability

The data that support the findings of this study are available from the corresponding author: J.A., upon reasonable request.

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
