# Peer review of "SiO2-Ag Composite as a Highly Virucidal Material: A Roadmap that Rapidly Eliminates SARS-CoV-2"

_nanomaterials, 2021, doi:10.3390/nano11030638_

Round 1

Reviewer 1 Report

The author reports a high virucide material constructed from SiO2-Ag composite immobilized in a polymeric matrix (ethyl vinyl acetate) that exhibited the high antibacterial activity towards Escherichia coli (E. coli) and Staphylococcus aureus (S. aureus) as well as towards SARS-CoV-2. Although, the work is of high quality and the manuscript is well written, a minor revision in the manuscript is necessary in order to publish the manuscript in Nanomaterials.
Comments: I am not convinced with an explanation of hot electrons over SiO2-Ag composite. How is it possible to transfer a charge between an insulator SiO2 and metal Ag ?

Author Response

Reviewer: 1

Comments:

The author reports a high virucide material constructed from SiO2-Ag composite immobilized in a polymeric matrix (ethyl vinyl acetate) that exhibited the high antibacterial activity towards Escherichia coli (E. coli) and Staphylococcus aureus (S. aureus) as well as towards SARS-CoV-2. Although, the work is of high quality and the manuscript is well written, a minor revision in the manuscript is necessary in order to publish the manuscript in Nanomaterials.

I am not convinced with an explanation of hot electrons over SiO2-Ag composite. How is it possible to transfer a charge between an insulator SiO2 and metal Ag?

Answer: In the present case, we have amorphous silica SiO2, composed of ordered tetrahedra which are randomly arranged throughout the material and can be considered a semiconductor with a band gap around 1.81 eV. This situation of different to the alpha-quartz SiO2 with a band gap of 9.0 eV, an insulating material (see the answer to the comment 4 of the reviewer 3). The mechanism to explain the charge transfer process between the semiconductor SiO2 and Ag is presented in the revised version of the manuscript (see paragraph 3, page 13).

Reviewer 2 Report

This is a very interesting paper that suggests new technologies in relation to bacterial/virucide purposes. In particular, the paper describes the synthesis and characterisation of a SiO2-Ag composite with antibacterial properties.  A wide range of characterisation techniques have been carried out, including Raman, TGA, FTIR, DRS, SEM, etc. The material displays very good antibacterial activity towards E. coli, S. aureus, and SARS-CoV-2.  Theoretical calculations have been performed in order to explain the mechanism. All the experimental and theoretical data have been discussed in detail. This is an interesting piece of work, the findings are very well supported by the experimental data, and considering also the fact that it has the potential to lead to new technologies, I am happy to recommend its publication in Nanomaterials. A minor comment: it would be great if the authors could compare the currently used antibacterial materials (in terms of cost and efficiency) with the one proposed in this paper.

Author Response

Reviewer: 2

Comments:

This is a very interesting paper that suggests new technologies in relation to bacterial/virucide purposes. In particular, the paper describes the synthesis and characterisation of a SiO2-Ag composite with antibacterial properties.  A wide range of characterisation techniques have been carried out, including Raman, TGA, FTIR, DRS, SEM, etc. The material displays very good antibacterial activity towards E. coli, S. aureus, and SARS-CoV-2. Theoretical calculations have been performed in order to explain the mechanism. All the experimental and theoretical data have been discussed in detail. This is an interesting piece of work, the findings are very well supported by the experimental data, and considering also the fact that it has the potential to lead to new technologies, I am happy to recommend its publication in Nanomaterials.

A minor comment: it would be great if the authors could compare the currently used antibacterial materials (in terms of cost and efficiency) with the one proposed in this paper.

Answer: In the reported articles at the bibliography, the minimum inhibitory concentration (MIC) value is usually employed as signature of biocide activity, which is obtained by varying the mass of a biocidal material against a given microorganism. In the present, the microbicidal tests are carried out from the contact surface of the solutions containing the microorganisms and the composite SiO2-Ag. Then, we were unable to compare with other antibacterial materials. In our case, we are capable to eliminate the SARS-CoV-2 (99.05%) by surface contact in 2 min (see the answer to the comment 2 of the reviewer 3). Our material is composed by amorphous SiO2 (99%), very cheap, and only 1% of expensive metallic Ag.

Reviewer 3 Report

In this study, the authors prepared a SiO2-Ag composite for the potential elimination of bacteria and SARS-CoV 2. In general, the article can be considered for publication after a major revision process. 

1) Chemical analysis of SiO2 and Ag should be tested to reveal the Si to Ag ratio (Ag wt%). In addition, it will be interesting to see the uniformity of Ag distribution over the selected area (EDX elemental mapping). 

2) Calculate the MIC of the prepared Eva-SiO2-Ag composite and compare this data with the literature.  

3) Supply interaction pictures (SEM or TEM) of composite with E.coli and S.aureus taken after some period of time. Describe the role of Eva. 

4) Is this composite is degradable with time (how long it can be effective)? Corresponding data should be shown.   

5) SiO2 is a well-known insulator, not sure why the authors stated it as n-type semiconductor. Degradation kinetics should be shown. Also, did the authors exclude the surface dye adsorption here? 

6) The authors used SiO2-Ag but not discussed recent advances in this field. Thus, recent studies dealing with SiO2-Ag antibacterial materials should be discussed. The following materials are the most efficient to date: 

- Cetyltrimethylammonium Bromide (CTAB)-loaded SiO2–Ag mesoporous nanocomposite as an efficient antibacterial agent. Nanomaterials 2021, 11, 477. 
- Boosting antibacterial activity with mesoporous silica nanoparticles supported silver nanoclusters. J. Colloid Interface Sci. 2019, 555, 470–479.  

Author Response

Reviewer: 3

Comments:

In this study, the authors prepared a SiO2-Ag composite for the potential elimination of bacteria and SARS-CoV 2. In general, the article can be considered for publication after a major revision process.

1) Chemical analysis of SiO2 and Ag should be tested to reveal the Si to Ag ratio (Ag wt%). In addition, it will be interesting to see the uniformity of Ag distribution over the selected area (EDX elemental mapping).

Answer: Following the reviewer’s suggestion, we have carried out punctual EDX on SiO2-Ag. These results are now included and discussed in the supporting material (see page 9, paragraph 1).

2) Calculate the MIC of the prepared Eva-SiO2-Ag composite and compare this data with the literature.

Answer: The final material is obtained in large plates by thermoplastic injection, with the SiO2-Ag particles incorporated in the EVA as polymeric matrix. The microbicidal tests are carried out from the contact surface of the solutions containing the microorganisms and EVA-SiO2-Ag, following ISO 22196 - Measurement of antibacterial activity on plastics and other non-porous surfaces and ISO 21702 - Measures of antiviral activity on plastics and other non-porous surfaces. Then we were unable to perform MIC tests because the SiO2-Ag concentration was fixed in the EVA, and we do not have a powder to be suspended with the microorganism solution and vary the mass in the solution (see the answer to the comment of the reviewer 2).

3) Supply interaction pictures (SEM or TEM) of composite with E.coli and S.aureus taken after some period of time. Describe the role of Eva.

Answer:  We were unable to do SEM or TEM images in a timely manner for reviewing the article. EVA is a flexible and low-cost polymer matrix used in different technological applications. In particular, as it is the focus of present work, SiO2-Ag composite present a high virucide activity against SARS-CoV-2. EVA becomes the engineering material, acting as a carrier of SiO2-Ag particles. A comment in the revised version of the manuscript is included (see page 5, paragraph 3).

4) Is this composite is degradable with time (how long it can be effective)? Corresponding data should be shown.

Answer: Bactericidal measurements of the material were performed after forced aging by ultraviolet irradiation, following ISO 4892-2: 2013 Plastics - Methods of exposure to laboratory light sources - Part 2: Xenon-arc lamps, which aims to reproduce the effects of weathering (temperature, humidity and/or wetting) that occur when materials are exposed in real-life environments to daylight or daylight filtered through window glass. It was observed that after simulating two years of aging (1200 hours of exposure) there is still a 99.950% reduction in the elimination of S. aureus ATCC 6538 and E. coli ATCC 8739. Thus, the durability used for the polymer was defined as two years. A new sentence in the revised version of the manuscript is now included to clarify this point (see page 12, paragraph 2).

5) SiO2 is a well-known insulator, not sure why the authors stated it as n-type semiconductor. Degradation kinetics should be shown. Also, did the authors exclude the surface dye adsorption here?

Answer: In the present case SiO2 is amorphous, then it works as a n-type semiconductor (see the answer of the comment to the reviewer 2).

The degradation kinetics of SiO2-Ag composite was added to the supplementary material. (see supplementary material, page 2, Figure SI2-A)

The surface adsorption of the dye in the SiO2-Ag composite was considered, and to reach the adsorptive equilibrium with the dye, it was left for 30 min in contact with RhB in the dark. Thus, no relevant RhB adsorption on SiO2-Ag was observed. The representation of Cn/C0 as function of time is now included in the supplementary material to clarify this point. (see supplementary material, page 2, Figure SI2-A)

6) The authors used SiO2-Ag but not discussed recent advances in this field. Thus, recent studies dealing with SiO2-Ag antibacterial materials should be discussed. The following materials are the most efficient to date:

- Cetyltrimethylammonium Bromide (CTAB)-loaded SiO2–Ag mesoporous nanocomposite as an efficient antibacterial agent. Nanomaterials 2021, 11, 477.

- Boosting antibacterial activity with mesoporous silica nanoparticles supported silver nanoclusters. J. Colloid Interface Sci. 2019, 555, 470–479.

Answer: We are grateful for the reviewer's suggestion. In the revised version of the manuscript, these works are cited and discussed. (see page 2, paragraph 4).

Round 2

Reviewer 3 Report

No more comments